# Bile Acid Diarrhea: From Molecular Mechanisms to Clinical Diagnosis and Treatment in the Era of Precision Medicine

**DOI:** 10.3390/ijms25031544

**Published:** 2024-01-26

**Authors:** Daiyu Yang, Chengzhen Lyu, Kun He, Ke Pang, Ziqi Guo, Dong Wu

**Affiliations:** 1Peking Union Medical College Hospital, Chinese Academy of Medical Sciences and Peking Union Medical College, Beijing 100730, China; yangdy20@student.pumc.edu.cn (D.Y.); pangk19@student.pumc.edu.cn (K.P.); guoziqi18@163.com (Z.G.); 2State Key Laboratory of Complex Severe and Rare Diseases, Department of Gastroenterology, Peking Union Medical College Hospital, Chinese Academy of Medical Sciences and Peking Union Medical College, Beijing 100730, China; chengzhenlyu@outlook.com (C.L.); hk6290418@163.com (K.H.)

**Keywords:** bile acid, diarrhea, farnesoid X receptor, fibroblast growth factor receptor 4, takeda G protein-coupled receptor 5, diagnosis, treatment

## Abstract

Bile acid diarrhea (BAD) is a multifaceted intestinal disorder involving intricate molecular mechanisms, including farnesoid X receptor (FXR), fibroblast growth factor receptor 4 (FGFR4), and Takeda G protein–coupled receptor 5 (TGR5). Current diagnostic methods encompass bile acid sequestrants (BAS), 48-h fecal bile acid tests, serum 7α-hydroxy-4-cholesten-3-one (C4), fibroblast growth factor 19 (FGF19) testing, and ^75^Selenium HomotauroCholic acid test (^75^SeHCAT). Treatment primarily involves BAS and FXR agonists. However, due to the limited sensitivity and specificity of current diagnostic methods, as well as suboptimal treatment efficacy and the presence of side effects, there is an urgent need to establish new diagnostic and treatment methods. While prior literature has summarized various diagnostic and treatment methods and the pathogenesis of BAD, no previous work has linked the two. This review offers a molecular perspective on the clinical diagnosis and treatment of BAD, with a focus on FXR, FGFR4, and TGR5, emphasizing the potential for identifying additional molecular mechanisms as treatment targets and bridging the gap between diagnostic and treatment methods and molecular mechanisms for a novel approach to the clinical management of BAD.

## 1. Introduction

Bile acids are synthesized by the liver and subsequently secreted into the gallbladder before traversing the intestine until reaching the terminal ileum, where they undergo recycling and re-entry into the liver, thereby establishing hepatoenteric circulation. Bile acids play pivotal physiological roles in human metabolism encompassing fat digestion, absorption of fat-soluble vitamins [1], as well as drug assimilation.

However, bile acids can also induce adverse reactions, including the development of bile acid diarrhea (BAD). BAD refers to the dysregulated recycling of bile acids within the enterohepatic circulation, characterized by either excessive biosynthesis/secretion or malabsorption of bile acids by the ileum. The presence of unabsorbed bile acids in the colon is believed to induce diarrhea through various mechanisms, including stimulation of fluid, mucus, or sodium secretion; enhancement of gastrointestinal motility; mucosal damage; and promotion of defecation [2]. The prevalence of BAD is relatively high. According to reports from developed countries, approximately 5% of the population experiences chronic diarrhea lasting more than four weeks at any given time; among these patients, about 25% are diagnosed with BAD. Therefore, it is estimated that approximately 1% of adults in Western countries suffer from BAD [3,4,5,6]. Among individuals diagnosed with irritable bowel syndrome (IBS), around 30% have an underlying condition of BAD [7]. A Danish study conducted between 2003 and 2021 employed ^75^SeHCAT testing referrals followed by a prescription of a bile acid sequestrant (colestyramine, colestipol, or colesevelam) within one year to identify the population affected by BAD. The study revealed regional variations across Denmark, with a total of 5264 individuals diagnosed with BAD. Compared to age- and sex-matched controls from the general population, the cohort with BAD exhibited a higher prevalence of comorbidities, increased healthcare utilization, as well as lower levels of education and income [8]. Unfortunately, data on the prevalence of IBS in eastern countries are currently unavailable. Although the prevalence of IBS in China is approximately 6.5% [9,10], the precise prevalence rate for BAD remains unknown. Regrettably, clinical recognition and diagnosis of BAD often receive insufficient attention.

The advent of the precision medicine era offers a novel approach to address the challenges in diagnosing and treating BAD. Precision medicine employs molecular mechanisms for individualized disease diagnosis and treatment. In the realm of BAD research, unraveling molecular mechanisms aids in comprehending disease pathogenesis and identifying new diagnostic markers and therapeutic targets. For instance, studying the synthesis, transport, and metabolism of bile acids at the molecular level can reveal new diagnostic indicators and treatment targets. This is the first review to link the molecular mechanisms of BAD with diagnostic and therapeutic approaches, providing a novel molecular perspective for the clinical management of BAD.

## 2. Pathogenesis and Clinical Manifestations

Three subtypes of BAD have been described: Type 1, patients with terminal ileal disease (e.g., Crohn’s disease, resection) or radiation injury resulting in impaired reabsorption of bile acids; type 2, idiopathic or primary; and type 3, other conditions (e.g., celiac disease, cholecystectomy) that alter intestinal motility or bile acid absorption [2]. The occurrence of BAD is attributed to the excessive delivery of bile acids to the colon, which can result from malabsorption, heightened hepatic synthesis, or unsynchronized delivery to the small intestine following cholecystectomy [11]. Patients with BAD exhibit enhanced colonic transit, heightened intestinal or colonic mucosal permeability, and an altered composition of the stool microbiome associated with diminished conversion of primary to secondary bile acids [12]. Although not universally applicable, several pathophysiologic mechanisms may arise due to the influence of bile acids in the colon (Figure 1).

### 2.1. FXR

The Farnesoid X receptor (FXR) plays a crucial role in regulating bile acid metabolism [13,14,15,16,17]. Under normal physiological conditions, bile acids are reabsorbed within the ileum and activate the FXR signaling pathway, thereby significantly promoting fibroblast growth factor 19 (FGF19) biosynthesis [18,19]. Subsequently, this growth factor is transported via portal venous circulation to hepatocytes where it effectively inhibits cholesterol 7α hydroxylase (CYP7A1) expression—an essential enzyme that controls hepatic bile acid synthesis rates [20]. FGF19 released into the portal circulation exerts an inhibitory effect on bile acid synthesis by blocking CYP7A1 through the activation of extracellular signal-regulated kinase (ERK)/c-Jun N-terminal kinase (JNK) [21]. Decreased secretion levels of FGF19 from ileal cells lead to attenuated suppression of CYP7A1 activity resulting in elevated endogenous production of bile acids. Consequently, inadequate absorption capacity for excessive amounts of these compounds within the small intestinal tract may contribute to diarrheal symptoms.

It is worth noting that although the regulation of CYP7A1 activity by FGF19 plays a crucial role, it should be recognized that the observed variation in FGF19 levels and disease severity could also be affected by other genetic variations affecting bile acid production [22]. In patients who developed secondary BAD after ileal resection surgery, Lenicek et al. demonstrated an association between the rs3808607 T>G polymorphism located within the promoter region of CYP7A1 and elevated 7α-hydroxy-4-cholesten-3-one (C4)-to-cholesterol ratios [23]. This finding suggests a two-fold increase in bile acid synthesis among individuals carrying the TT (AA) genotype. Likewise, another study investigating CYP7A1 polymorphisms about both fecal primary bile acid levels and colorectal adenoma risk revealed that individuals heterozygous for the rs8192877 A>G variant exhibited approximately twice as high fecal primary bile acids compared to those homozygous for AA [24]; moreover, GG homozygotes displayed even higher values.

Other postulated mechanisms may account for the triglyceride-lowering effect of FXR activation. Activation of FXR modulates free fatty acid (FFA) oxidation and enhances triglyceride clearance [16,17]. Incubation of human hepatoma HepG2 cells with FXR ligands, such as chenodeoxycholic acid (CDCA), and agonists have demonstrated the induction of peroxisome proliferator-activated receptor α (PPARα) expression and its target genes to promote FFA oxidation. PPARα is believed to play a crucial role in mediating triglyceride metabolism. Fibrates activate PPARα, leading to reduced hepatic apoC-Ⅲ production and increased LPL-mediated lipolysis, resulting in enhanced catabolism of triglyceride-rich particles and decreased secretion of very low-density lipoproteins (VLDLs), thereby causing hypotriglyceridemia. Currently, there is limited clinical data on triglyceride metabolism in patients with BAD. Emerging evidence suggests that approximately one-third of patients with BAD exhibit hypertriglyceridemia. Further research is necessary to establish the risk factors associated with hypertriglyceridemia in patients with BAD and elucidate the underlying mechanisms, which would enable targeted treatment [25].

### 2.2. FGFR4

FGF19, synthesized by ileal enterocytes in response to bile acids binding to FXR, interacts with fibroblast growth factor receptor 4 (FGFR4) expressed on hepatocyte cell membranes [26]. Through enterohepatic circulation, FGF19 binds to and activates hepatic FGFR4 in complex with the transmembrane co-receptor Klothoβ (KLB), thereby initiating an intracellular signal pathway associated with ERK1/2 activation [27]. This subsequently leads to the feedback inhibition of CYP7A1 expression and suppression of bile acid synthesis. Additionally, the presence of genetic polymorphisms in FGFR4 and KLB also contributes to variations in FGF19 response, which have been reported to demonstrate significant associations with primary BAD and IBS-D [28,29].

One of the compelling lines of evidence supporting the involvement of FGFR4 in bile acid regulation arises from the phenotypic assessment of FGFR4^−/−^ mice, which exhibit a diminished gallbladder size and a two- to three-fold elevation in secreted bile acid levels compared to their wild-type counterparts. Furthermore, disruption of the FGFR4 gene leads to a seven-fold increase in liver HMG-CoA reductase activity, the rate-limiting enzyme in cholesterol biosynthesis, as well as a 2.5-fold upregulation of CYP7A1, responsible for converting cholesterol into the bile acid precursor 7a-hydroxycholesterol [30]. Hence, it is evident that abrogation of FGFR4’s negative control on cholesterol metabolism towards bile acid synthesis occurs upon disruption of the FGFR4 gene. Additionally, FGFR4^−/−^ mice demonstrate significantly elevated expression levels of cytochrome P450 family 8 subfamily B member 1 (CYP8B1), an essential component for cholic acid (CA) -derived but not CDCA-derived bile acid production [31]. Consequently, basal levels of cholic acid and its metabolites relative to CDCA are increased in these mice lacking functional FGFR4. The indispensable role played by FGFR4 specifically and not other members within the FGFR family in FGF19-mediated regulation of bile acids has been further substantiated through in vivo experiments utilizing an engineered variant specific to activating only FGFR4 while still downregulating CYP7A1 expression within hepatic tissue [32]. Notably, the impaired release of FGF19 from the ileum has been observed in cholestyra-mine-responsive diarrhea-predominant irritable bowel syndrome (IBS-D), indicating excessive hepatic bile acid synthesis due to disrupted FGF19 signaling as a cause for bile acid-induced diarrhea [33].

### 2.3. TGR5

The pathogenesis of BAD involves disruptions in the bile acid pathways, resulting in excessive delivery of primary bile acids to the colon. Takeda G protein–coupled receptor 5 (TGR5), also known as G protein-coupled bile acid receptor 1 (GPBAR1), is expressed on enteric nerves and enterochromaffin cells, and its activation can regulate motility in the small intestine and colon [1]. Upon reaching the colonic lumen, primary or secondary bile acids activate TGR5 on the cell membrane of colonocytes, promoting colonic motility, fluid, and electrolyte secretion, increased mucosal permeability, and visceral sensitization that ultimately leads to diarrhea. In the study conducted by Alemi [34], analysis of gastrointestinal and colon transit, defecation frequency, water content, contractility, peristalsis, and transmitter release was performed in wild-type (*tgr5-wt*), knockout (*tgr5-ko*), and transgenic mice (*tgr5-tg*) to investigate the role of the TGR5 receptor in colonic motility. The findings suggest that TGR5 mediates the effects of bile acids on colonic motility; deficiency of TGR5 leads to constipation in mice. Therefore, targeting TGR5 may hold therapeutic potential for digestive diseases. All types of bile acids have a binding affinity for TGR5; however, lithocholic acid (LCA) exhibits particularly potent activity [1,35,36]. In the study conducted by Wei [37], TGR5 immunoreactivity was predominantly observed in the crypts of recto-sigmoid biopsies and also detected in scattered cells within the lamina propria. Notably, TGR5 expression was significantly elevated in IBS-D patients compared to controls, unlike the expression of the control vitamin D receptor. Moreover, TGR5 expression exhibited a positive correlation with pain scores and a positive association with primary bile acids (CA and CDCA), while displaying a negative correlation with secondary bile acids (deoxycholic acids (DCA) and LCA).

Additionally, a study conducted by Camilleri [38] revealed that genetic variation in TGR5 may contribute to alterations in small bowel transit (SBT) and colonic transit. The association between the TGR5 SNP rs11554825 (minor allele frequency 41%) and symptom phenotype (total cohort), as well as intermediate phenotype (SBT or colonic transit assessed through radioscintigraphy), was investigated in a sample comprising 230 healthy controls and 414 patients with lower functional gastrointestinal disorders (FGID), including IBS-alternators [Alt] (*n* = 84), IBS-constipation [IBS-C] (*n* = 157), and IBS-D (*n* = 173). Further studies are required to elucidate the potential role of TGR5 in the mechanism and treatment of bowel dysfunction.

### 2.4. An Altered Gut Microbiome

Patients with BAD exhibit increased intestinal permeability and an altered gut microbiome, characterized by significantly lower microbial diversity and a distinct stool bacterial composition compared to patients without BAD. One analysis encompassed a total of 257 samples obtained from 134 patients. Notably, patients with BAD exhibited a significant reduction in α-diversity [39]. The gut microbiome plays a crucial role in regulating the size and composition of the bile acids pool, which in turn affects the characteristics of the microbiome. Dysbiosis has been observed in patients with IBS, particularly IBS-D, where there is an increase in *Escherichia coli*, *Bacteroides*, and *Bifidobacterium* [40]. Studies have also shown that IBS-D patients have an elevated relative abundance of *Firmicutes* such as *Ruminococcaceae spp*. and *Clostridium cluster XIVa* but reduced levels of *Bacteroides* [41,42]. Furthermore, some authors reported that 25% of IBS-D patients had higher levels of *Clostridia* bacteria like *C. scindens* [33,43]. Patients with BAD exhibited significantly reduced alpha diversity and distinct compositional profiles based on beta diversity compared to patients with IBS-D without abnormal bile acid metabolism (ABAM). At the phylum, genus, and species levels, patients with BAD and IBS-D without ABAM displayed divergent microbiome compositions. Specifically, patients with BAD demonstrated a higher *Firmicutes* to *Bacteroidetes* ratio in comparison to those with IBS-D without ABAM. Notably, there were 29 differentially abundant bacterial genera between the two groups; of these, 26 genera showed decreased abundance in BAD including *Alistipes*, *Clostridium*, and *Bacteroides*. Moreover, 70 differentially abundant bacterial species were identified between the two groups; among these species, 61 exhibited decreased abundance in BAD. The diminished species included various *Clostridia* such as *phoceensis*, *polynesiense,* and *leptum*; *Faecalibacterium prausnitzii*; as well as *Alistipes obesi* and *Alistipes finegoldii*. Conversely, *Erysipelatoclostridium ramosum* showed increased abundance [7,44].

In addition to the overall association between diagnosed BAD and increased alpha and beta diversity of the microbiota, as well as a higher *Firmicutes* to *Bacteroidetes* ratio, significant associations were observed between fasting serum C4 levels and primary fecal bile acids in stool with alpha and beta diversity. Furthermore, both biochemical indices exhibited correlations with differential taxa diversity in patients with BAD. Functional analysis revealed notable differences in bacterial-encoded enzymes within microbial species linked to BAD, particularly a decreased expression of bile acid thiol ligase according to the Kyoto Encyclopedia of Genes and Genomes (KEGG) pathway analysis [7]. These enzymes play a crucial role in converting primary bile acids into secondary bile acids and are associated with the observed higher percentage of primary bile acids detected in stool samples from patients with BAD. 

The symptoms of BAD typically present as chronic diarrhea. However, based on the available data, the clinical practice guideline [2] recommends utilizing a patient’s medical history of terminal ileal resection, cholecystectomy, or radiotherapy rather than solely relying on symptom presentation during the initial assessment to aid in identifying patients with BAD. Patients diagnosed with BAD experience a diminished quality of life, characterized by heightened concerns regarding loss of bowel control and an increased emphasis on proximity to restroom facilities compared to patients with IBS-D. These symptoms have been closely associated with depression, as assessed using the Hospital Anxiety and Depression Questionnaire [45]. Treatment of BAD using bile acid sequestrants (BAS) has shown significant improvements in mean scores for the “Role limitation due to physical health” dimension and overall mental component summary, as measured by the 36-Item Short Form Survey (SF-36) [46].

These debilitating symptoms significantly impact patients’ quality of life; however, due to their resemblance to other gastrointestinal disorders, misdiagnosis or oversight is prevalent. A comprehensive understanding of the pathogenesis and clinical manifestations of BAD holds paramount importance for its accurate diagnosis and effective treatment.

## 3. Diagnosis of BAD

In this section, we will discuss the diagnostic methods for BAD, which can be categorized into two types: existing diagnostic methods and potential alternative diagnostic methods based on molecular mechanisms (Table 1). Regarding the existing diagnostic methods, we will classify them according to their underlying mechanisms, including bile acids detection, diagnostic tests utilizing BAS, and molecular biomarkers detection based on bile acid metabolism.

### 3.1. Existing Diagnostic Methods

Bile acids detection: Unraveling potential multi-mechanistic factors contributing to enhanced bile acid excretion in patients [49].

#### 3.1.1. ^75^Selenium HomotauroCholic Acid Test (^75^SeHCAT)

This diagnostic method is utilized for the assessment of bile acid retention in the body, specifically in the diagnosis of BAD [50,51], and is currently acknowledged as the gold standard for diagnosing BAD within the academic community. The procedure involves administering radiolabeled taurocholic acid followed by measuring the excretion rate of the radiolabeled isotope after 7 days to evaluate bile acid recovery. Diagnostic criteria state that “<5% 7-day WBR (whole-body retention) of ^75^SeHCAT indicates severe malabsorption; <10% indicates severe and moderate malabsorption; <15% indicates severe, moderate, and mild malabsorption.” [5] By providing quantitative data on bile acid retention in the body, ^75^SeHCAT enables more precise diagnosis. However, employing radiolabeled isotopes exposes patients to radiation, although test doses are generally considered safe. The method necessitates specialized facilities and equipment for monitoring and handling radioactive substances which restricts its widespread use and poses implementation challenges. 

#### 3.1.2. 48 h Fecal Bile Acid Test

The diagnosis of BAD can be achieved through a 48 h fecal bile acid test, which measures the content of bile acids in the last two days of feces following a 100 g fat diet for four days. Given the limited availability of ^75^SeHCAT in many regions, measuring fecal bile acids serves as a viable alternative [52,53]. There are three diagnostic criteria for BAD: total fecal bile acids ≥2337 μmol/48 h, primary bile acids >10%, or total fecal bile acids ≥1000 μmol/48 h combined with primary bile acids >4% [54]. A study has shown that the sensitivity and specificity of detecting ^75^SeHCAT <15% with primary bile acids >10% are 45% and 63% respectively, indicating that the diagnosis based solely on primary bile acids testing may not be very definitive. It is recommended to combine the measurement of total fecal bile acid content to make a more accurate judgment [55]. Typically, high-performance liquid chromatography (HPLC) or mass spectrometry is employed to measure fecal bile acids [56,57]. This method is relatively non-invasive as it only requires the collection of fecal samples from patients without the need for endoscopic examinations or exposure to radioactive substances. It is also straightforward and does not necessitate complex equipment or techniques. Moreover, collecting continuous fecal samples over a 48-h period provides a reasonably accurate reflection of the patient’s bile acid excretion during that time frame. However, this method can be time-consuming and inconvenient for patients while posing challenges for healthcare workers.

#### 3.1.3. BAS Diagnostic Therapy

BAS such as cholestyramine or colesevelam are selected for diagnostic therapy based on the patient’s clinical symptoms and medical history [58,59]. The efficacy of the treatment is evaluated by monitoring the patient’s symptoms and reactions. If there is a significant improvement in symptoms, such as resolution or reduction of diarrhea, it can serve as an indication for diagnosing BAD. This approach offers simplicity since it only requires medication intake and observation of treatment effectiveness to confirm BAD diagnosis. However, this method does not directly measure bile acid excretion but rather relies on inferring BAD possibility from the patient’s response to medication. Moreover, uncertainty exists regarding drug response due to potential influence from various factors, making it challenging to definitively determine whether symptom improvement results from BAD presence or could lead to adverse drug reactions.

#### 3.1.4. Molecular Biomarkers Detection

As described in the molecular mechanism, activation of FXR by bile acids stimulates the production of FGF-19, which inhibits C4 as a rate-limiting enzyme to reduce liver bile acid synthesis. Measurement of serum C4 and serum FGF19 levels enables the direct assessment of hepatic synthesis, secretion, and intestinal absorption of bile acids, and high-level C4 or low-level FGF19 can indicate the presence of BAD [60,61]. The detection of C4 and FGF19 has been validated by the gold standard ^75^SeHCAT [62,63]. Compared to 48-h fecal bile acid, the specificity of C4 and FGF19 are 83% and 78%, respectively, and the sensitivity of C4 and FGF19 is 29% (The diagnostic criteria for serum C4 and FGF19 are C4 > 52.5 ng/mL or FGF19 ≤ 61.7 pg/mL). This indicates that further improvements are needed in the diagnostic sensitivity of these two testing methods [64]. 

It is worth mentioning that current research has proposed a combination of fecal bile acid measurements and serum C4 measurements for diagnostic purposes [3,65]. A recent study revealed that the combination of C4 detection with ≥1.1 daily watery stools (Bristol type 6 and 7) had 70% (51–85%) sensitivity and 95% (83–99%) specificity, the logistic regression model including C4, the sum of measured stool bile acids and daily watery stools, had 77% (58–90%) sensitivity and 93% (80–98%) specificity, suggesting considerable improvements in the sensitivity of C4-based testing [66]. However, there is an urgent need to establish diagnostic criteria, which necessitates further in-depth research. For instance, large-scale clinical studies are required to establish unified diagnostic criteria for BAD by assessing levels of fecal bile acid and serum C4. Moreover, optimization of measurement methods for fecal bile acid and serum C4, including sampling times, is necessary to enhance their accuracy and repeatability. Additionally, machine learning techniques and other approaches can be employed to develop models predicting the occurrence and severity of BAD based on results from fecal bile acid and serum C4 measurements with support from database algorithms [66,67].

### 3.2. Potential Diagnostic Methods

#### 3.2.1. Other Potential Molecular Biomarkers Detection

The binding of bile acids to TGR5 and FXR represents a potential diagnostic approach for BAD. Activation of these receptors by bile acids initiates a cascade of signaling pathways that impact bile acid metabolism and other physiological functions [68,69]. However, the effects induced by different bile acids binding to a specific receptor can vary [1,70]. Therefore, quantifying the concentrations of various bile acids and utilizing molecular imaging techniques to determine the ratio of different bile acids bound to receptors can assess the activation levels of TGR5 and FXR signaling pathways, thereby identifying abnormalities in BA binding to TGR5 and FXR in patients with BAD. Nevertheless, further research and validation are necessary for the clinical implementation of these methods to ascertain their diagnostic value and feasibility.

In addition, a recent study on the serum lipidome found that the detection of decanoylcarnitine, cholesterol ester (22:5), eicosatrienoic acid, L-alpha-lysophosphatidylinositol (18:0), and phosphatidylethanolamine (O-16:0/18:1) could distinguish BAD from controls with a sensitivity of 78% (64–89%) and a specificity of 93% (83–98%) [67]. Further research on this test is needed.

#### 3.2.2. Detecting Intestinal Microbiota and Its Metabolites

The intestinal microbiota plays a pivotal role in bile acid metabolism and regulation [36]. Also, BAs are important determinants of the abundance, diversity, and metabolic activity of the microbiota [71]. In patients with BAD, there may exist significant disparities in the composition and metabolites of the intestinal microbiota compared to healthy individuals. These differences can encompass variations in microbial types, abundance, as well as metabolite types and levels. Such discrepancies are closely associated with the occurrence and progression of BAD, highlighting the involvement of the intestinal microbiota in disrupting bile acid metabolism. For instance, elevated levels of primary bile acids during early development result in the proliferation of bacteria that express genes involved in bile acid metabolism within the small intestine [72]. In recent years, a series of studies have proposed novel methods, such as the diagnosis of BAD through the detection of volatile organic compounds (VOCs) produced by gut microbiota metabolism. By analyzing the VOCs, the status of the gut microbiota can be inferred, allowing for the assessment of bile acid metabolism and subsequent diagnosis of BAD [73,74,75,76]. By conducting bacteriome analysis and volatile gas testing on both BAD patients and healthy individuals, it is possible to identify distinguishing features. These distinctions can be utilized to establish specific diagnostic criteria and methods for identifying BAD patients by setting thresholds for particular microbiota structures and metabolite levels that differentiate them from healthy individuals. This approach holds promise for non-invasive early diagnosis and real-time monitoring of BAD; however, it necessitates complex experimental procedures and data analysis. Moreover, this diagnostic method may encounter challenges related to standardization, calibration issues, as well as inter-individual variability.

## 4. Treatment of BAD

### 4.1. Existing Treatment Methods

Existing treatment modalities primarily encompass diagnostic intervention employing BAS and therapeutic administration of FXR agonists.

BAS are oral medications that form non-absorbable complexes by binding to unbound bile acids in the intestine, thereby reducing intestinal reabsorption of bile acids and their stimulation of the intestine, ultimately alleviating symptoms associated with diarrhea. However, BAS has side effects and poor patient compliance, and long-term use may affect the absorption of fat-soluble vitamins [77]. Commonly used BAS include cholestyramine, colesevelam, and colestipol [78]. A randomized trial found that colesevelam increases the delivery of total and secondary BAs to stool, hepatic BA synthesis, and colonic mucosal expression of genes that regulate BA, farnesoid X, and GPBAR1 receptors [79]. Another clinical trial indicated that colesevelam was superior to placebo at inducing remission of BAD diagnosed with C4 concentration greater than 46 ng/mL [80]. Further experience with this study is awaited.

FXR agonists modulate bile acid synthesis, secretion, and metabolism by inhibiting the rate-limiting enzyme involved in hepatic bile acid production through the aforementioned molecular mechanism, thereby influencing bile acid levels and ameliorating symptoms of bile acid disorders. Currently, FXR agonists such as obeticholic acid (OCA) are employed for BAD treatment [81,82]. Furthermore, a randomized clinical trial has demonstrated that tropifexor increased FGF19 levels and decreased C4 levels for a duration of up to 8 h. No significant differences in stool frequency, or stool form between treatments [47].

### 4.2. Potential Treatment Methods

There may exist potential diagnostic methodologies and treatment targets that are not encompassed by the current diagnostic modalities.

As mentioned in the preceding text regarding potential diagnostic methods, molecules such as TGR5 (which exhibits distinct responses to various bile acids), FGF19, and C4 play pivotal roles in the pathogenesis and progression of BAD. Consequently, these molecules hold promise as prospective therapeutic targets. In a recent randomized, double-blinded, placebo-controlled (1:1 ratio) trial, it was observed that aldafermin, a 190 amino acid engineered peptide analog of human FGF19 with 95.4% homology, led to a significant reduction in serum C4, fecal total bile acids, and the percentage of secretory bile acids at both days 14 and 28 [48].

In the exploration of novel treatment methods, the intestinal microbiota can serve as a promising starting point. Recently, there has been significant attention on the association between the intestinal microbiota and human health [83]. Studies have demonstrated that imbalances in the intestinal microbiota may be implicated in the onset and progression of BAD or other conditions such as liver disease [71]. Consequently, modulation of the intestinal microbiota holds potential as a novel therapeutic approach for BAD management. Furthermore, specific bacterial species within the intestinal microbiota are involved in bile acid metabolism and regulation of intestinal epithelial cell function [84], rendering them prospective targets for intervention. A multicenter study utilizing 16S-sequencing has established a link between the presence of endoscopically visible mucosal biofilms and a dysbiotic gut microbiome. Additionally, metabolomic analysis revealed an accumulation of bile acids within biofilms that correlated with fecal bile acid excretion, thereby linking this phenotype with a mechanism of diarrhea [85]. Manipulation of these bacterial species’ abundance and activity could potentially ameliorate symptoms experienced by patients with BAD. Additionally, metabolites produced by the intestinal microbiota may impact bile acid metabolism and absorption within the intestines; thus, regulating levels of these metabolites presents another plausible avenue for treating BAD.

## 5. Significance and Limitation

This is the first review to integrate the molecular mechanisms of BAD with diagnostic and therapeutic methods, facilitating the development of personalized and effective strategies for the diagnosis and treatment of BAD. Furthermore, it provides other researchers with a molecular perspective on the diagnosis and treatment of BAD. 

However, due to the current incomplete understanding of the molecular mechanisms behind BAD, the scope of this review may be constrained by the limitations of existing literature. Furthermore, due to individual variability and the complexity of the disease, generalizing the results of specific molecular studies to a broader population with BAD may be challenging.

## 6. Conclusions

BAD is a prevalent intestinal disease characterized by the multifaceted involvement of FXR, FGFR4, and TGR5 in its pathogenesis. Bile acids play a pivotal regulatory role in the intestine, participating in vital processes such as fat digestion and absorption, as well as cholesterol metabolism. Current diagnostic approaches primarily encompass treatment with BAS, 48-h fecal bile acid test, serum C4 and FGF19 testing, and ^75^SeHCAT; while treatment strategies mainly revolve around BAS administration and FXR agonists. However, despite the existing diagnostic and therapeutic modalities available for BAD management, there are still challenges to be addressed in improving diagnosis accuracy and treatment efficacy. By analyzing the existing diagnostic and therapeutic methods for BAD and their molecular mechanisms, we believe that this review can provide subsequent researchers with a molecular perspective on the diagnosis and treatment of BAD.

## 7. Perspective

This review presents a novel perspective on the clinical diagnosis and treatment of BAD by exploring its molecular mechanisms, which can serve as a reference for future researchers to develop precision medicine strategies at the molecular level. In addition to the established diagnostic and therapeutic methods, other potential targets based on molecular mechanisms could be explored. Moreover, intestinal microbiota plays a crucial role in the development of BAD by modulating bile acid metabolism and regulating intestinal epithelial cell function. Given the current limitations in diagnostic and therapeutic methods for BAD, future research should focus on elucidating its underlying molecular mechanisms and their interactions with gut microbiota to facilitate more accurate diagnosis and personalized treatment. Such efforts will provide a comprehensive theoretical basis for precision medicine while offering new insights into developing innovative diagnostic tools and therapies.

## Figures and Tables

**Figure 1 ijms-25-01544-f001:**
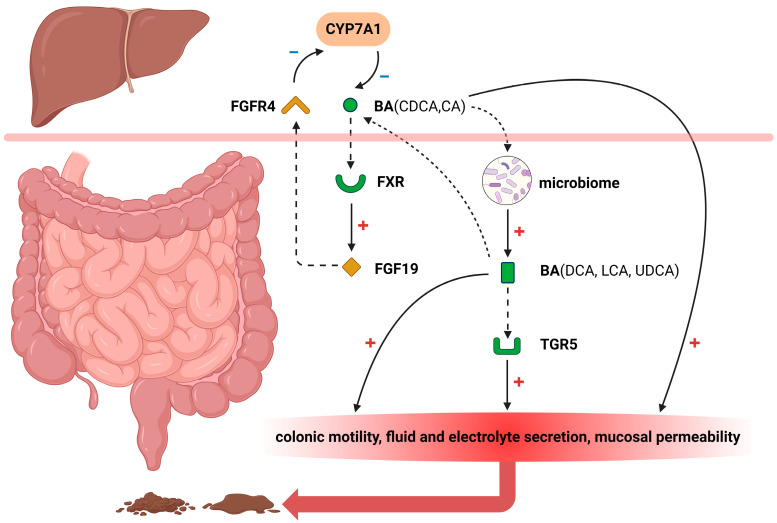
Bile acids, such as chenodeoxycholic acid (CDCA) and cholic acid (CA), are transported from the liver to the gut postprandially. Subsequently, these bile acids activate the farnesoid X receptor (FXR), leading to the production of fibroblast growth factor 19 (FGF19) which is then transported back to the liver. FGF19 binds to fibroblast growth factor receptor 4 (FGFR4) and modulates cholesterol 7α hydroxylase (CYP7A1), thereby influencing bile acid synthesis in hepatocytes. After being released into the gut by the liver, primary bile acids can be converted into secondary bile acids including deoxycholic acid (DCA), lithocholic acid (LCA), and ursodeoxycholic acid (UDCA) through microbial metabolism. These secondary bile acids are passively absorbed and subsequently recycled back to the liver via portal circulation for reutilization. Alternatively, they may interact with Takeda G protein-coupled receptor 5(TGR5). Moreover, the excessive accumulation of diverse bile acids in the gut can also stimulate colonic motility, fluid, and electrolyte secretion, and enhance mucosal permeability, ultimately leading to diarrhea—a cardinal manifestation of BAD. Created by Biorender.com.

**Table 1 ijms-25-01544-t001:** Current bile acid diarrhea (BAD) diagnostic tests and potential treatment methods are based on the underlying mechanisms of diagnostic tests [3,47,48].

Diagnostic Test	^75^SeHCAT	48 h Fecal Bile Acid Test	BAS Diagnostic Therapy	Serum C4	Serum FGF19
Molecular mechanism	Radionuclide-labeled reagents; Simulate bile acid metabolism	Dietary manipulation replicates bile acid metabolism.	Drugs chelate excess bile acids.	C4 is a rate-limiting enzyme in the synthesis of bile acids.	FGF19, FXR, FGFR4 facilitate endogenous bile acid synthesis.
Diagnostic criteria	<5% (severe) <10% (moderate) <15% (mild)	Total fecal Bas ≥ 2337 μmol/48 h; OR Primary BA > 4% + total fecal BA > 1000 μmol/48 h OR Primary BA> 10%	Based on the patient’s clinical symptoms	>52.5 ng/mL	<61.7 pg/mL
Advantage	Reliable; High Sensitivity and Specificity	Quantitative assessment of BAs; No radiation	Non-invasive test; widely used; No radiation	relatively simple; No radiation	No radiation; commercial ELISA assay
Disadvantage	Expensive; Not be widely available; Radiation exposure	Collection Challenges; Interpretation Complexity	No clear diagnosis; Potential Side Effects; Poor compliance	Non-specificity; Variation; Confirmation required	Confirmation required
Potential treatment method	Reduce the generation and excretion of bile acids in the body			FXR agonists; Anti-C4 drugs	FXR agonists; FGF19 analog

## Data Availability

Not applicable.

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
