# Peer review of "Bile Acid Diarrhea: From Molecular Mechanisms to Clinical Diagnosis and Treatment in the Era of Precision Medicine"

_ijms, 2024, doi:10.3390/ijms25031544_

Round 1

Reviewer 1 Report

Comments and Suggestions for Authors

This manuscript aims to be a much-needed review of molecular mechanisms in bile acid diarrhea and how these may be employed in diagnosis and treatment.  This is an important topic to update as the disorder is poorly recognised and often undiagnosed.  Unfortunately this review seems to be based on other review articles and has omitted many specific primary papers describing individual mechanisms, particularly some of those recently published.  It appears that the authors have little clinical or research experience of BAD.  The article would be improved by revision to correct some imbalance in the topics, and to include papers publishing recent evidence of mechanisms.

Specific points:

page 1, line 40.  Ref. 2 is misplaced here.  There are better introductions to define BAD.

page 1, line 42.  This prevalence figure is uncited and unclear what it refers to.  Is it in the population, or in IBS-D / functional diarrhea?  See ref 39, or 6.

page 2, line 46.  The 1% population prevalence was first described in ref 39.

page 2. line 56.  Please cite references to the Chinese data.

page 3.  It would be appropriate to cite differences in CYP7A1 and bile acid production before describing regulation by FGF19 (see PMID: 35526572).  There are clinical data on KLB and FGFR4 variants as well.  The clinical evidence and primary studies showing low FGF19 in BAD is more relevant than the mouse studies described here and on p4.

page 4.  TGR5.  Although potentially of interest, ref 25 is the only clinically relevant work on this and BAD.  Perhaps this should be omitted as a focus in the Abstract and elsewhere.

page 5, line 196.  Please check that the references are correctly applied to the microbiota findings.  Ref 31 I think is wrong here.

page 5, lines 208-9.  The manifestation of BAD is incorrect and not supported by the reference given.  Please refer to some of the many correct descriptions of the clinical presentation and problems in BAD.  

page 5-7, table 1.  This would be better in a different format.  The potential treatments referenced in the heading are out of place here, and are not all described later.

page 8, line 289.  7a-OH-4-cholesten-3-one has been mentioned before and has been abbreviated to 7aC4 and now C4. Please make consistent

page 8, lines 289 -303.  This section omits the initial papers on these markers and an important, very recent, assessment (PMID: 37794830).  Lines 296 - 301 repeat earlier statements.

page 8, lines 308 -314 should refer to the recent paper PMID: 37794830 and particularly to PMID: 37072179.

page 9, lines 336-46.  More detail would help here.

page 9, line 249 is unclear.

page 9, line 360.  There is only a little published evidence supporting the use of FXR agonists.  This refers to obeticholic acid but there is also a paper on tropifexor (PMID: 32702169).

page 9, lines 375-380.  Further speculation could cite PMID: 34146566 and others (https://doi.org/10.3390/microbiolres12020023).

Comments on the Quality of English Language

The quality of English is good and does not need significant editing.

Reviewer 2 Report

Comments and Suggestions for Authors

This manuscript is well well-written and comprehensive review dealing update of bile acid diarrhea (BAD), which is interesting in this area and also in related diseases such as constipation and IBS.

Mechanisms of BAD including farnesoid X receptor (FXR), fibroblast growth factor receptor 4 (FGFR4), and takeda G protein–coupled receptor 5 (TGR5) is reviewed comprehensively but, in order to help readers understand appropriate schematic drawings are needed in this section.

Current diagnostic methods of bile acid diarrhea including bile acid sequestrants (BAS), 48-hour fecal bile acid tests, serum 7α-hydroxy-4-cholesten-3-one (C4), fibro- 18 blast growth factor 19 (FGF19) testing, and 75 Selenium HomotauroCholic Acid Test (75SeHCAT) are presented properly and if it is possible schematic drawings are warranted here. Lastly, representative treatment using BAS and FXR agonists are explained shortly.

Comments on the Quality of English Language

English is without problem.
